# A NEW PARADIGM FOR CROSS-MODALITY PERSON RE-IDENTIFICATION

## ABSTRACT

Visible and infrared Person Re-identification(ReID) is still very challenging on account of few cross-modality dataset and large inter-modality variation. Most existing cross-modality ReID methods have trouble eliminating cross-modality discrepancy resulting from the heterogeneous images. In this paper, we present an effective framework and build a large benchmark, named NPU-ReID. To this end, we propose a dual-path fusion network and taking transformer as the smallest feature extraction unit. To expand cross-modality sample diversity, we propose a modality augmentation strategy to generate semi-modality pedestrian images by exchanging certain patch and the main innovation is that the cross-modality gap can be indirectly minimized by reducing the variance of semi-modality and infrared or visible modality. Moreover, in order to make the traditional triplet loss more suitable for cross-modal matching tasks, multi-masking triplet loss is a targeted design for optimizing the relative distance between anchor and positive/negative samples pairs from cross-modality, especially constraining the distance between simple and hard positive samples. Experimental results demonstrate that our proposed method achieves superior performance than other methods on SYSU-MM01, RegDB and our proposed NPU-ReID dataset, especially on the RegDB dataset with significant improvement of 6.81% in rank1 and 9.65% in mAP.

## 1 INTRODUCTION

Person re-identification (ReID) is a challenging task in computer vision, which is widely used in autonomous driving, intelligent video surveillance and human-computer interaction systems Ye et al. (2021); Zheng et al. (2019); Miao et al. (2019). Person ReID aims to search target pedestrian across multiple non-overlapping surveillance cameras or from different video clips. At present, most researches performed on single-modality visible images captured in daytime has achieved good performance, such as TransReID He et al. (2021), AGW Ye et al. (2021), MMT Ge et al. (2020), HOReID Wang et al. (2020), PAT Li et al. (2021) and ISP Zhu et al. (2020). However, in night-time surveillance and low-light environments, visible cameras fail to capture person images with rich appearance information. The light limitation determines that single-modality ReID framework fails to satisfy all-weather practical application scenarios.

With cameras which can be switched to infrared mode being widely used in intelligent surveillance systems, cross-modality infrared-visible ReID has been a key but challenging technology. Visible images and infrared images are heterogeneous images pairs with very different visual features. Intuitively, pedestrians in visible images have clearer texture features and valid appearance information than infrared images under good illumination environment, and infrared images can provide more distinct pedestrian outward appearance and integrated contour information. Naturally, robust features representation can be generated by sufficiently incorporating cross-modality complementary information. However, single modality person ReID method is difficult to be directly used for cross-modality tasks because of large inter-modality variations. The differences of images belonging to the same identity from cross-modality will be even greater than that of images belonging to the different identity from the same modality. The large modality gap between visible images and infrared images and unknown environmental factors arises a vitally challenging cross-modality problem. As is shown in Figure 1, the same identity from the same modality suffers from large intra-modality variations arising from different human poses as well as diverse camera viewpoints. Meanwhile, the

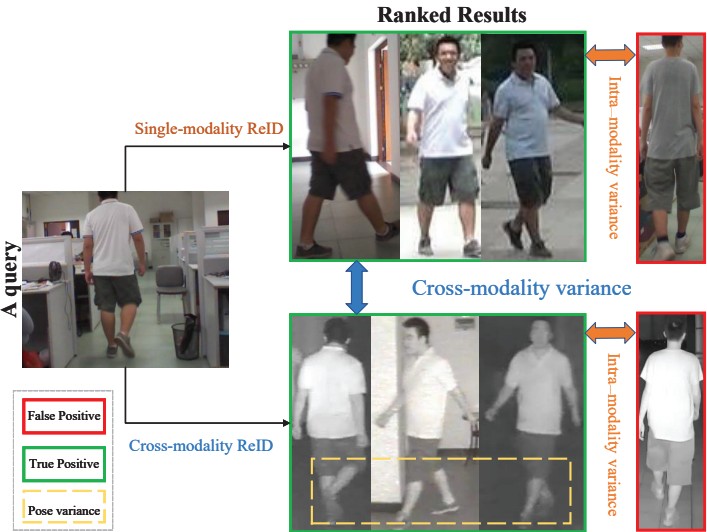

Figure 1: Illustration of the Person Re-Identification. When the left visible image is used as a query, the images list above is ranked results in single-modality and the images list below is ranked results in cross-modality. The 1st to 3rd cols present True Positive samples, which means these gallery images and the query images all belong to the same person. The last col presents False Positive samples.

heterogeneous imaging processes of different spectrum cameras result in large cross-modality variations. These variations may lead to larger intra-identification difference than cross-identification difference, and then cause wrong matching results. Therefore, it is demanded to prompt a solution to reduce the cross-modality discrepancy and intra-modality variations.

Researchers have proposed many methods to address the aforementioned challenges in cross-modality ReID. Several methods map persons images from different modality into a common feature space to minimize modality gap Ye et al. (2018a;b; 2020;?). To alleviate the color discrepancy, generative adversarial networks (GANs) is used to synthesize fake RGB/IR images while preserving the identity information as much as possible in many works Wang et al. (2019; 2020); Wang et al. (2019); Zhang et al. (2019). However, there is still the challenge of appearance variations including background clutter and viewpoint variations. Furthermore, these methods continue to use triplet loss or ranking loss in single-modality metric learning to supervise the network to mining identity-related cues, rather than designing modality-related loss function to learn discriminative features in the cross-modality setting.

The quality of dataset directly affect the representation ability of embedding feature, which determines the accuracy and efficiency of identification to some extent. Consequently, we build a cross-modality dataset called NPU-ReID, which makes up for the deficiency of small-scale and uneven modality distribution. We collect images with multi-view camera system consisting of 4 visible cameras and 4 infrared cameras, which ensures that each identity has several infrared and visible images under each camera. Aiming at tackling the concurrent challenge in intra- and cross-modality variations, we present a novel modality augmentation to to eliminate the modality discrepancy. The straightforward operation is to generate semi-modality images by exchanging certain regions with a patch from images of the same identity from another modality, which can deepen the information communication between infrared images and visible images. Simply, the augmented image contains two types of information from different modality, which can effectively reduce the difficulty of cross-modal matching.

In addition, the ReID network always trained with cross entropy loss and triplet loss to improve the discrepancy between inter-category and intra-category. We propose the Multi-masking triplet loss to neutralize advantages and disadvantages of traditional triplet loss and triplet loss with hard sample mining and design a cross-modality positive sample distance compress function to reduce intra-category difference.

The main contributions of this paper are as follows:

- We build a new paradigm for cross-modality person re-identification, which outperforms state-of-the-arts on SYSU-MM01, RegDB and our proposed NPU-ReID dataset, especially on the RegDB dataset with significant improvement of 6.81%/9.65% on rank1/mAP 91.84%.

- We build NPU-ReID, a comprehensive visible-infrared dataset for person re-identification tasks, giving in total 34621 visible images and 38578 infrared images of 600 identities.

- We present a modality augmentation strategy to fully exploit heterogeneous image properties, which is plug-and-play and can be easily applied to the most existing methods for cross-modality tasks. To guide network to learn more powerful and generic features, We design a multi-mask triplet loss for cross-modality recognition to compress the distance between anchor from one modality and corresponding sample pairs from the other modality.

## 2 RELATED WORKS

### 2.1 CROSS-MODALITY PERSON RE-IDENTIFICATION

Cross-modality Person re-identification has developed rapidly Ye et al. (2020); Wang et al. (2019); Li et al. (2020); Wang et al. (2019); Chen et al. (2021), which has achieved higher and higher accuracy in RegDB Nguyen et al. (2017) and SYSU-MM01 Wu et al. (2017). Generally, in neural networks, two-stream structure Ye et al. (2020); Ye et al. (2018a) and FC layer structure are used to extract the features with different modalities separately and calculate loss through the distance metric. Li et al. (2020) introduced X-modality to the shared feature space, which was generated by a lightweight network including two convolution layers and one ReLU layer on visible images. Ye et al. Ye et al. (2021) proposed a channel augmentation strategy. The primary idea is to evenly generate color independent images by randomly exchanging channels of RGB. Thanks to the widespread application of GANs in recent years, several method leverage GANs to apply cross-modality style transfer, feature disentanglement or intermediate modality generation Wang et al. (2020); Choi et al. (2020); Wang et al. (2019).

The comparison among existing commonly used Re-ID datasets is shown in Table 1. Single modality person re-identification has made great achievements supported by constantly emerging visible ReID datasets, such as Market 1501 Zheng et al. (2015), DukeMTMC Zheng et al. (2017), CUHK03 Li et al. (2014) and MSMT17 Wei et al. (2018). There are two datasets for visible and thermal Person re-identification tasks, RegDB Nguyen et al. (2017) and SYSU-MM01 Wu et al. (2017). The images in SYSU-MM01 dataset are collected by 6 cameras including two infrared cameras and four visible cameras.The scale of RegDB dataset is smaller than that of SYSU-MM01 dataset, including 4120 infrared images and 4120 visible images, corresponding to 412 identities.

Recent years have witnessed remarkable progress in Face Recognition and RGB ReID task, with a variety of approaches proposed in the literatures and applied in real applications. Liu et al. Liu & Tan (2021) proposed hetero-center triplet loss on the basis of HC loss Zhu et al. (2020). Hetero-center triplet loss improve HC loss in the form of triplet, which constrain the distance of different class centers from both intra-modality and cross-modality and reduce the number of calculations. Lv et al. Lv et al. (2022) proposed the whole constraint and partial triplet-center loss (WCPTL) to deal with the discrepancy between different modalities in local features. But center loss with triplet is so keen to coincidence of centers of the same ID from cross-modality that decrease the inter-category discrepancy.

## 3 THE NPU-REID DATASET

### 3.1 DATASET DESCRIPTION

We propose a cross-modality multi-view Re-ID dataset named NPU-ReID. In this section, we will talk about how we collect and annotate images, and then analyze the advantages, disadvantages and application scenarios of the dataset.

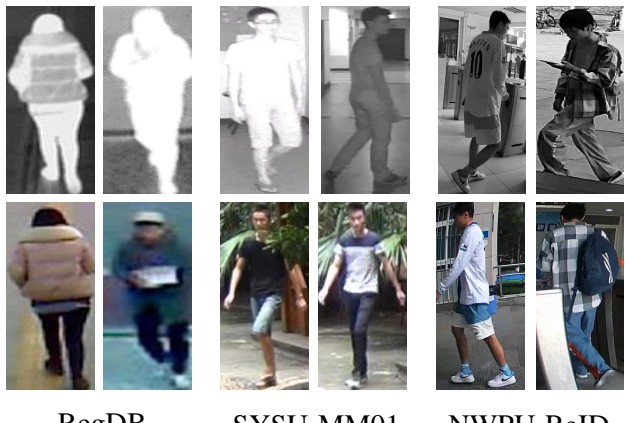

RegDB          SYSU-MM01          NWPU-ReID

Figure 2: Examples of RGB images and thermal images in SYSU-MM01 dataset, RegDB dataset and our NPU-ReID. Each columns are of the different person. In each image pairs, the images on the top is infrared and the images on the bottom is visible.

Since deep learning performance benefits from the dataset that it is trained on, it is inevitable to build a large-scale and comprehensive dataset. For this purpose, we present a new dataset named NPU-ReID for visible-thermal Person re-identification, covering 600 identities respectively. NPU-ReID dataset contains images captured by 8 cameras, including four visible cameras and four thermal cameras. To ensure that the number of pedestrian images of every identity is as equal as possible and there are peoples with different poses and viewpoints in images, visible cameras and thermal cameras are mutually crosswise installed. Moreover, there are also equal numbers of cameras installed indoor or outdoor. Since the thermal images have only one channel, they are expanded to three channels at the same value so that feed them into the network with 3-channel visible images.

There are 600 valid pedestrians identities in NPU-ReID dataset. We have a fixed split using 480 identities for training, 120 for validation and 120 for testing. During training, all images of the 480 persons in training dataset in 8 cameras can be sampled. During testing, there are two test modes, Visible to Thermal or Thermal to Visible. Visible to Thermal mode means that the images from thermal modality were used as the gallery set while those from visible modality (default is visible) were used as the query set. In both modes, we adopt three forms of the scope of search and two ways to select gallery dataset, which motivated by SYSU-MM01. For the scope of search, indoor-search, outdoor-search and all-search all can be selected and details In indoor-search mode, the gallery set only contains the visible images captured by four indoor visible cameras. Generally, the all-search mode is more challenging than the indoor-search mode. The single-shot and multi-shot setting are main two ways to select gallery dataset, where 1 or 10 images of a person are randomly selected to form the gallery set. Significantly, since each of the eight cameras has a different view, only the images under the camera in which query is sampled need to be skipped during the search.

Table 1: Comparison of NPU-ReID and exsiting ReID datasets. There are datasets for single modality and cross-modality ReID. IP denotes the number of images per identity.

| Dataset | ID | Visible | Thermal | cameras | IP |
|---|---|---|---|---|---|
| Single modality datasets | | | | | |
| Market-1501 | 1501 | 32,668 | 0 | 6 | 21.76 |
| CUHK03 | 1467 | 13164 | 0 | 6 | 8.97 |
| MSMT17 | 4101 | 126441 | 0 | 15 | 30.83 |
| DukeMTMC-reID | 1404 | 36441 | 0 | 8 | 25.95 |
| Cross-modality datasets | | | | | |
| RegDB | 412 | 4120 | 4120 | 2 | 20.00 |
| SYSU-MM01 | 491 | 30071 | 15792 | 6 | 93.40 |
| NPU-ReID | 600 | 34621 | 38578 | 8 | 122.00 |

## 3.2 SPECIALITY.

Some of the visualizations results of NPU-ReID, RegDB and SYSU-MM01 dataset are shown in Figure 2. The images from two modalities in RegDB dataset are captured by visible and thermal cameras installed in the same location, so there are no viewpoint change in the pedestrian images of the same identity, which fails to satisfy the task of matching pedestrian images across camera views. In SYSU-MM01 dataset, the number of visible images is several times that of infrared images, that is the problem of modality imbalance, which fails to support Visible to Thermal mode. However, in NPU-ReID, the images captured by different cameras differs greatly with different viewpoints and poses. The view angle of the dataset differs greatly and the number of images in each modality is balanced compared to others. As can be seen from Figure 2, images in our dataset has a higher resolution and contains more characteristic information of pedestrians.

## 4 OUR PROPOSED METHOD

In this section, we introduce the overview framework of our proposed cross-modality ReID method. Our framework mainly consists of three components: modality augmentation, the dual-path representation learning network based on transformer and metric learning with identity loss and our proposed multi-mask triplet loss.

## 4.1 OVERALL ARCHITECTURE

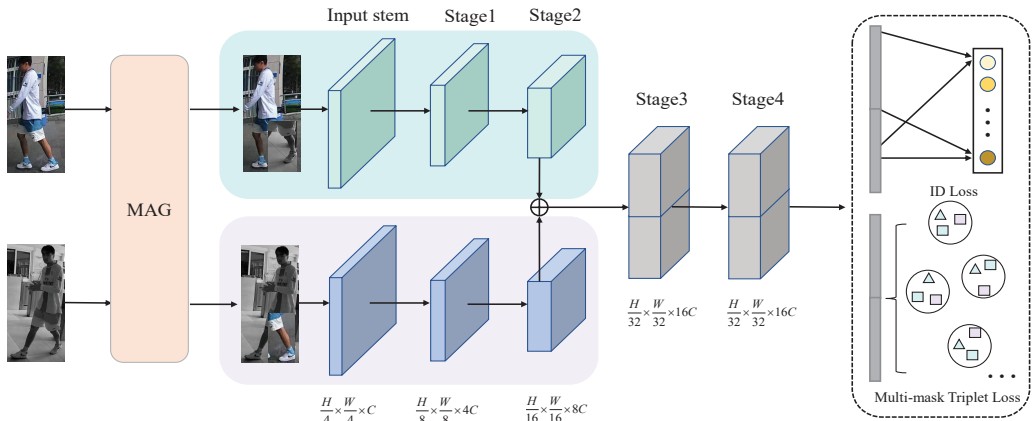

Figure 3: Our proposed framework for cross-modality ReID.

The transformer-based methods are springing up like mushrooms, but there is no specific model designed for cross-modality person re-identification. We propose a carefully-designed dual-path transformer network for infrared-visible person re-identification, as is shown in Figure 3. We explore how many parameters a two-stream network based on transformer should share, which is still not well investigated in the existing literature. The network consists of input stem module and four stages. The first two stages form the modality-specific branch and the following two stages form the modality-shared branch.

Given two image of size H×W, we first feed them into modality augmentation(MAG) to operate data transforms. Then the augmented images are resized to 224×224 and split into non-overlapping patches of size P×P to reduce computational complexity. We have verified that taking the image of size 224×224 as input can achieve better performance than the original image through several experiments, which demonstrates that square images are more helpful for window partition, window merging and attention calculation.

In stage 1 and stage 2, the transformer blocks are used to capture infrared features and visible features in parallel simultaneously. The design of the transformer block is motivated by the Swin Transformer's success Liu et al. (2021), which is built with encoder, bottleneck and decoder. The in-

put patches are fed into the transformer block following patch merging layer to generate hierarchical features with multi-scale and decrease the computational complexity with respect to image size.

In order to avoid the lost of a large amount of specific-modality feature information due to concentrating on learning cross-modality shared features, specific-modality features are learned in the first two stages respectively, and shared-modality features are learned in the last two stages. The advantages of this framework have been proved in A.3. Finally, all features are used to calculate identity loss and multi-masking triplet loss to optimize the identity relationship among different person from cross-modality.

In this section, we first analyze the impact of triplet loss functions with different variants concerning of samples pairs selection and weight contribution of hard samples on model performance.

Commonly, the sample strategy random sample $P$ identities within each batch, of which $K$ visible and $K$ infrared images are random selected. In triplet loss, all selected simple pairs are considered equally for the overall loss calculation, which limits network's ability to identify more informative pairs among the selected ones. Whereas triplet loss with hard example mining only consider hard samples for each triplet, it is the other extreme form compared triplet loss. This method abandons a majority of samples and fails to take full advantage of all train samples, resulting at the training is inefficient.

## 4.2 THE MULTI-MASKING TRIPLET LOSS

Through abundant experiments, we find that the positive samples are all from another modality and the negative samples are almost from the same modality for anchor sample from one modality.The former is undoubted because the variance in heterogeneous images is larger than that in homogeneous images of the same identity. The latter makes cross-modality retrieval degenerate into single-modality retrieval, raising the problem of mis-alignment in task. In training parse, anchor and negative samples are all from the same modality, while query and gallery are from two different modality in reference. Therefore, the network should attach more importance to learning discriminative features from different modalities in cross-modality ReID.

Moreover, we also find that the hard positive samples of all anchors from one identities are always the same sample from another modality, which indicates that hard examples are probably outliers. These outliers may magnify intra-category variance and exist stably even when the model is converged.

Figure 4: Illustration of the hard triplet pairs. The left circle contains visible images of category I and infrared images of category II, and the right circle contains infrared images of category I and visible images of category II. The dotted line points to the corresponding hard positive samples.

Therefore, to solve the problem mentioned above, in addition to the baseline identity loss with cross entropy loss, we adopt a moderate cross-modality triplet loss function, multi-masking triplet(MMT) loss. The loss, consisting of MMAP and MMAN loss, is used to optimize the distance of cross-modality positive pairs and negative pairs. The loss inherits the strength of relative distance optimization from common triplet loss and meanwhile relieves the problem of mis-align task and outlier sample, as is shown in Figure 4.

Specifically, for visible anchor sample, we select negative samples only in thermal images. In calculation. The number of negative sample pairs to be selected is reduced from $2(P-1)K$ to $(P-1)K$. Then we reduce the inter-modality discrepancy by constraining the distance between the hard sample and the easy sample from thermal modality as shown in Figure 5. Anchor and negative sample from different modalities are pushed far away and positive sample from different modalities are pulled closer, which is consistent to our analysis. As Visible to Thermal mode an example, the loss function is defined as:

$$L_{mtri} = \sum_{a \in batch} \left[ \left( \max_{p \in V} d_{a,p} - \min_{n \in T} d_{a,n} + m_1 \right) + \lambda \left( \max_{p \in V} d_{a,p} - \min_{p \in V} d_{a,p} - m_2 \right) \right] \quad (1)$$

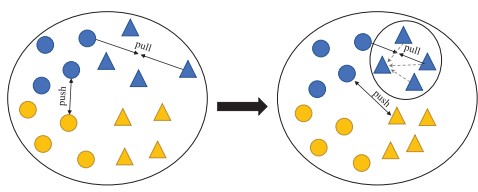

where $d_{a,p/n}$ denotes the Euclidean distance of anchor samples and positive/negtive samples. $V$ and $T$ denote visible modality and thermal modality.

Our design has two benefits: 1) It is likely to consider all the possible triplet relations in the augmented image set. We can take two different samples from the same modality into account at the same time, which is beneficial to reduce the intra-category difference and improve the recognition accuracy. 2) Compared with the method which constrains the distance between one sample and the center of its category by center loss, our method avoids that the outlier sample destroy other pairwise distance.

Figure 5: Illustration of the proposed MMT Loss. Graphics with different colors denote features belonging to different identities and the dot and the triangle means features from visible modality and infrared modality respectively.

### 4.3 THE MODALITY AUGMENTATION

For cross-modality visible-thermal person re-identification, one of the challenge is to learn fused feature embedding of the same identity from different modalities. The cross-modality matching is usually formulated by learning modality shared or invariant features. Our image augmentation strategy is similar to cutmix, while the critical difference is that the removed regions are filled with patches from the image with the same identity from another modality rather than random training images. Some visualization results of our augmentation method are shown in Figure 6.

We describe the MAG operation in detail. The visible training set and the thermal training set of the identity $i$ are denoted as $V^i = \{v_1^i, v_2^i, \cdots, v_k^i\}$ and $T^i = \{t_1^i, t_2^i, \cdots, t_k^i\}$. The goal of MAG is to generate new training images $v_{k^i}'$ and $t_{k^i}'$ by exchanging part region of $v_k^i$ and $t_k^i$. A pair of images is exchanged partly by the probability of $p$, which can be used to simulate the image of obscured pedestrian. The sample strategy of exchanging region $(x_1, y_1, x_2, y_2)$ is as follows. Assume the size of input image is $H \times W$, then the aera of origin image is $S = H \times W$. We constrain the area ratio of the exchanging region to the original image to $[A_l, A_h]$ and the aspect ratio of exchanging region is randomly initialized to $[r_l, r_h]$. Then, we can infer that the height and width of the exchanging region is $h = \sqrt{S \times A \times r}, w = \sqrt{\frac{S \times A}{r}}$ by calculation. Finally, we can determine the size, shape and position of the exchanging region by randomly initializing a center point $c = (x_c, y_c)$.

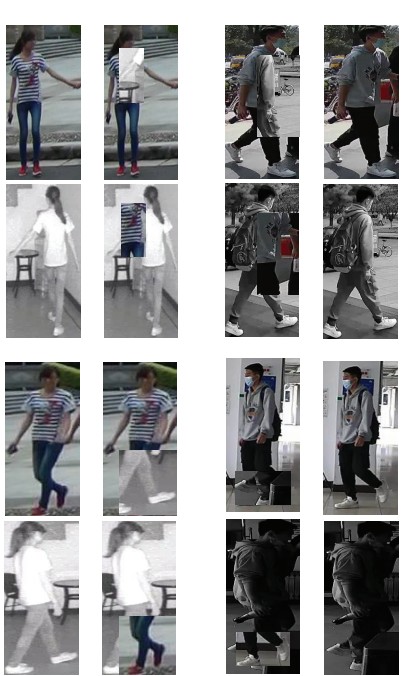

Original Images    Region Exchanged Augmentation    Original Images

Figure 6: Illustration of the area exchangeable augmentation in visible-infrared person re-identification, where the images in first column and last column original images from RegDB and NPU-ReID. The middle two columns display augmented images.

The basic motivation behind the Modality Augmentation (MAG) is that the cross-modality gap can be indirectly minimized by reducing the variance of semi-modality and infrared or visible modality. Since the labels of the original image are the same as those of the exchanging area, the labels of the augmented image remain unchanged. It can let model pay attention to the efficient discriminative information buried in the shared features simultaneously increase training efficiency. Moreover, the MAG method also can be regarded as a special regional pixel-level image fusion. The fused image not only contains the visible feature which represents the color information of the appearance, but also contains the thermal feature which represents the contour information. More importantly, the augmented

Table 2: Evaluation of MAG and MMT on three cross-modality ReID datasets, where the 'B' refers to the most two-path IV-ReID network training the network only with cross-entropy loss.

| | Method | | | NPU-ReID | | | | RegDB | | | | SYSU-MM01 | | | |
|---|---|---|---|---|---|---|---|---|---|---|---|---|---|---|---|
| B | MAG | MMTN | MMTP | R1 | R10 | R20 | mAP | R1 | R10 | R20 | mAP | R1 | R10 | R20 | mAP |
| ✓ | | | | 88.66 | 97.26 | 99.15 | 79.27 | 86.89 | 95.73 | 98.12 | 83.91 | 52.49 | 91.18 | 96.14 | 54.19 |
| ✓ | ✓ | | | 90.1 | 98.14 | 99.37 | 82.11 | 89.05 | 96.74 | 98.37 | 85.68 | 58.91 | 92.27 | 96.71 | 56.33 |
| ✓ | | ✓ | | 92.87 | 98.52 | 99.79 | 82.98 | 91.19 | 96.82 | 98.58 | 86.43 | 61.76 | 92.58 | 96.56 | 58.14 |
| ✓ | | ✓ | ✓ | 93.35 | 98.88 | 99.63 | 83.61 | 91.74 | 97.07 | 98.71 | 87.94 | 63.35 | 93.02 | 96.73 | 58.92 |
| ✓ | ✓ | ✓ | ✓ | **94.41** | **99.21** | **99.9** | **84.92** | **91.84** | **98.01** | **98.93** | **88.79** | **64.16** | **93.42** | **97.5** | **60.17** |

images from two different modalities corresponding to the same ID label, which can guarantee that the intra-class distance of the feature mapped to the shared space is shorter than the inter-modality distance. Therefore, the intra-ID inter-modality variant can be effectively reduced. We observe that the MAG successfully lets the model less focus on discriminative but modality-specific features and more focus on common but modality-shared features. Our design has two benefits: 1) The area exchanging strategy can replaced random region in original images with a patch in the image from the other modality by chance, which improves generalization and localization by letting a model pay attention to both the most discriminative parts of objects and the entire object region rather than excessive depending on certain region. 2) It can be seamlessly integrated with other common image augmentation operations and introduces a minimal computation overhead.

## 5 EXPERIMENTAL RESULTS

In this section, we analyze the effectiveness of each component in our our network and compare our results with existing state-of-the-art (SOTA) algorithms.

### 5.1 MODULE ABLATION

We evaluate the effect of the cross-modality augmentation strategy and the multi-masking triplet loss on all three ReID datasets and the results are shown in Table 2. We observe the performance on the three datasets improved significantly when applying our designed cross-modality augmentation operation. Our proposed MAG evenly improves the baseline rank-1 and mAP by about 4% and 3% on three datasets.When incorporating with the multi-masking triplet loss, the performance is further dramatically reinforced. Our proposed MMTN and MMTP work together and are mutually reinforcing, improving the baseline mAP by an average four percent. It can be noted that the overall performance reaches the best when utilizing the mixed loss function of MMTN and MMTP.

Table 3: Comparison with state-of-the-art methods on RegDB dataset.

| Method | Venue | Visible to Thermal | | | | Thermal to Visible | | | |
|---|---|---|---|---|---|---|---|---|---|
| | | R1 | R10 | R20 | mAP | R1 | R10 | R20 | mAP |
| BDTR | IJCAI 18 | 33.56 | 58.61 | 67.43 | 32.76 | 32.92 | 58.46 | 68.43 | 31.96 |
| Deep Zero-padding | ICCV 17 | 17.75 | 34.21 | 44.35 | 18.90 | 16.63 | 34.68 | 44.25 | 17.82 |
| MAC | MM 19 | 36.43 | 62.36 | 71.63 | 37.03 | 36.20 | 61.68 | 70.99 | 36.63 |
| MSR | TIP 19 | 48.43 | 70.32 | 79.95 | 48.67 | - | - | - | - |
| D2RL | CVPR 19 | 43.40 | 66.10 | 76.30 | 44.10 | - | - | - | - |
| AlignGAN | ICCV 19 | 57.90 | - | - | 53.60 | 56.30 | - | - | 53.40 |
| cm-SSFT | CVPR 20 | 72.30 | 0.00 | - | 72.90 | - | - | - | - |
| X-modality | AAAI 20 | 62.21 | 83.13 | 91.72 | 60.18 | - | - | - | - |
| DDAG | ECCV 20 | 69.34 | 86.19 | 91.49 | 63.46 | 68.06 | 85.15 | 90.31 | 61.80 |
| HAT | TIFS20 | 71.83 | 87.16 | 92.16 | 67.56 | - | - | - | - |
| Hi-CMD | CVPR 20 | 70.93 | 86.39 | - | 66.04 | - | - | - | - |
| JSIA-ReID | AAAI 20 | 48.50 | - | - | 49.30 | 48.10 | - | - | 48.90 |
| AGW | TPAMI 21 | 70.05 | - | - | 66.37 | - | - | - | - |
| NFS | CVPR 21 | 80.54 | 91.96 | 95.07 | 72.10 | 77.95 | 90.45 | 93.62 | 69.79 |
| CAJ | ICCV21 | 85.03 | 95.49 | 97.54 | 79.14 | 70.02 | 86.45 | 91.61 | 66.30 |
| Our method | - | **91.84** | **98.01** | **98.93** | **88.79** | **89.54** | **97.29** | **98.60** | **86.22** |

This subsection presents a comparison with the current state-of-the-art methods on RegDB, NPU-ReID and SYSU-MM01 datasets, as shown in Table 3, Table 4 and Table 5 respectively.

**Comparisons on RegDB dataset.** The experiments on the RegDB dataset demonstrate that the proposed method obtains the best performance under visible to infrared mode and infrared to visible

mode. Specially, we achieve rank-1 accuracy of 91.84% and mAP of 88.79% in Visible to Thermal mode, and rank-1 accuracy of 89.54% and mAP of 86.22% in thermal to visible mode and the mAP on RegDB climbed above 85 for the first time.

Table 4: Comparison with state-of-the-art on NPU-ReID dataset.

| Methods | Venue | All-Search | | | | | | | | Indoor-Search | | | | | | | |
|---|---|---|---|---|---|---|---|---|---|---|---|---|---|---|---|---|---|
| | | Visible to Thermal | | | | Thermal to Visible | | | | Visible to Thermal | | | | Thermal to Visible | | | |
| | | R1 | R10 | R20 | mAP | R1 | R10 | R20 | mAP | R1 | R10 | R20 | mAP | R1 | R10 | R20 | mAP |
| BDTR | TIP 20 | 66.73 | 84.37 | 92.18 | 47.21 | 65.29 | 83.23 | 91.98 | 47.03 | 54.38 | 83.77 | 90.53 | 49.02 | 55.21 | 83.21 | 90.48 | 49.15 |
| DDAG | ECCV 20 | 90.80 | 99.25 | 99.72 | 79.99 | 89.82 | 99.32 | 99.78 | 78.55 | 86.29 | 96.22 | 99.45 | 80.89 | 82.89 | 98.13 | 98.58 | 80.54 |
| AGW | TPAMI 21 | 90.97 | 99.13 | 99.60 | 80.96 | 90.10 | 99.38 | 99.84 | 79.67 | 88.87 | 97.13 | 99.11 | 71.76 | 73.96 | 97.56 | 99.02 | 70.94 |
| TSLFN | NC 21 | 90.27 | 99.13 | 99.70 | 78.84 | 90.68 | 99.30 | 99.69 | 79.99 | 90.19 | 98.89 | 99.34 | 79.10 | 89.86 | 98.62 | 99.56 | 80.21 |
| CAJ | ICCV21 | 70.49 | 82.43 | 98.16 | 54.59 | 68.59 | 93.64 | 98.35 | 54.74 | 53.97 | 83.36 | 92.22 | 50.83 | 54.27 | 83.51 | 91.98 | 52.19 |
| Ours | - | **94.41** | **99.41** | **99.90** | **84.92** | **93.46** | **99.08** | **99.89** | **83.57** | **85.91** | **98.76** | **99.75** | **85.90** | **83.37** | **98.10** | **99.67** | **84.22** |

Table 5: Comparison with the state-of-the-arts on SYSU-MM01.

| Methods | Venue | All-Search | | | | | | | | Indoor-Search | | | | | | | |
|---|---|---|---|---|---|---|---|---|---|---|---|---|---|---|---|---|---|
| | | Single-Shot | | | | Multi-Shot | | | | Single-Shot | | | | Multi-Shot | | | |
| | | R1 | R10 | R20 | mAP | R1 | R10 | R20 | mAP | R1 | R10 | R20 | mAP | R1 | R10 | R20 | mAP |
| HOG | CVPR05 | 2.74 | 18.91 | 32.51 | 4.28 | 3.25 | 21.82 | 36.51 | 2.04 | 4.38 | 29.96 | 50.43 | 8.7 | 4.62 | 34.22 | 56.28 | 3.87 |
| LOMO | CVPR15 | 1.75 | 14.14 | 26.63 | 3.48 | 1.96 | 15.06 | 27.3 | 1.85 | 2.24 | 22.53 | 41.53 | 6.64 | 2.24 | 22.79 | 41.8 | 3.31 |
| BDTR | IJCAI 18 | 17.01 | 55.43 | 71.96 | 19.66 | - | - | - | - | - | - | - | - | - | - | - | - |
| Deep Zero-padding | ICCV 17 | 14.8 | 54.12 | 71.33 | 15.95 | 19.13 | 61.4 | 78.41 | 10.89 | 20.58 | 68.38 | 85.79 | 26.92 | 24.43 | 75.86 | 91.32 | 18.64 |
| cmGAN | IJCAI 18 | 26.97 | 67.51 | 80.56 | 27.8 | 31.49 | 72.74 | 85.01 | 22.27 | 31.63 | 77.23 | 89.18 | 42.19 | 37 | 80.94 | 92.11 | 32.76 |
| D2RL | CVPR 19 | 28.9 | 70.6 | 82.4 | 29.2 | - | - | - | - | - | - | - | - | - | - | - | - |
| AlignGAN | ICCV 19 | 42.4 | 85 | 93.7 | 40.7 | 51.5 | 89.4 | 95.7 | 33.9 | 45.9 | 87.6 | 94.4 | 54.3 | 57.1 | 92.7 | 97.4 | 45.3 |
| MAC | MM 19 | 33.26 | 79.04 | 90.09 | 36.22 | - | - | - | - | - | - | - | - | 46.56 | 93.57 | 98.8 | 40.08 |
| MSR | TIP 19 | 37.35 | 83.40 | 93.34 | 38.11 | 43.86 | 86.94 | 95.68 | 30.48 | 39.64 | 89.29 | 97.66 | 50.88 | 53.05 | 93.71 | 98.93 | 47.48 |
| DEF | MM 19 | 48.71 | 88.86 | 95.27 | 48.59 | 54.63 | 91.62 | 96.83 | 42.14 | 52.25 | 89.86 | 95.85 | 59.68 | - | - | - | - |
| HPILN | IET IP 19 | 41.36 | 84.78 | 94.51 | 42.95 | 47.56 | 88.13 | 95.98 | 36.08 | 45.77 | 91.82 | 98.46 | 56.52 | - | - | - | - |
| cm-SSFT | CVPR 20 | 47.70 | 54.1 | - | - | 57.4 | 59.1 | - | - | - | - | - | - | 59.62 | 94.45 | 98.07 | 50.6 |
| X-modality | AAAI 20 | 49.92 | 89.79 | 95.96 | 50.73 | - | - | - | - | - | - | - | - | 60.42 | 96.88 | 99.5 | 53.52 |
| CMM+CML | MM 20 | 51.80 | 92.72 | 97.71 | 51.21 | 56.27 | 94.08 | 98.12 | 43.39 | 54.98 | 94.38 | 99.41 | 63.7 | - | - | - | - |
| DDAG | ECCV 20 | 54.75 | 90.39 | 95.81 | 53.02 | - | - | - | - | 61.02 | 94.06 | 98.41 | 67.98 | - | - | - | - |
| Hi-CMD | CVPR 20 | 34.94 | 77.58 | - | 35.94 | - | - | - | - | - | - | - | - | - | - | - | - |
| JSIA-ReID | AAAI 20 | 38.1 | 80.7 | 89.9 | 36.9 | 45.1 | 85.7 | 93.8 | 29.5 | 43.8 | 86.2 | 94.2 | 52.9 | 52.7 | 91.1 | 96.4 | 42.7 |
| TSLFN | NeuroC 21 | 56.96 | 91.5 | 96.82 | 54.95 | 62.09 | 93.74 | 97.85 | 48.02 | 59.74 | 92.07 | 96.22 | 64.91 | 69.76 | 95.85 | 98.9 | 57.81 |
| AGW | TPAMI 21 | 47.50 | 84.39 | 92.14 | 47.65 | - | - | - | - | 54.17 | 91.14 | 95.98 | 62.97 | - | - | - | - |
| HAT | TIFS 21 | 55.29 | 92.14 | 97.36 | 53.89 | - | - | - | - | 62.1 | 95.75 | 99.2 | 69.37 | 69.76 | 95.85 | 98.9 | 57.81 |
| NFS | CVPR 21 | 56.91 | 91.34 | 96.52 | 55.45 | 63.51 | 94.42 | 97.81 | 48.56 | 62.79 | 96.53 | 99.07 | 69.79 | 70.03 | 97.7 | 99.51 | 61.45 |
| Our method | - | **64.16** | **93.42** | **97.5** | **60.17** | **67.61** | **96.28** | **98.64** | **56.72** | **64.53** | **97.66** | **99.37** | **70.73** | **73.28** | **97.12** | **98.74** | **63.44** |

**Comparisons on NPU-ReID dataset.** To keep things fair, we evaluate the state-of-the-art VI-ReID methods published in the last years on our NPU-ReID dataset. We apply Visible to Thermal mode and Thermal to Visible mode in the case of single-shot in all-search and indoor-search testing mode respectively. It is worth mentioning that our proposed method achieves the best performance under various settings. In Visible to Thermal all-search mode, our method outperforms achieves 94.41% rank-1 and 84.92% mAP.

**Comparisons on SYSU-MM01 dataset.** The results on SYSU-MM01 are shown in Table 5, our method has outperformed the state-of-the-art methods. In single-shot all-search mode, our method achieves 64.16% Rank-1 and 60.17% mAP and it surpasses the performance of the state-of-the-art method.

Above results all demonstrate that our proposed model outperforms existing method. We have achieved much better performance under all-search and indoor-search settings, suggesting that our method can learn better intra-modality specific features and cross-modality sharing features by well designing modality augmentation strategy and multi-masking triplet loss.

# 6 CONCLUSION

In this paper, we propose NPU-ReID, a visible-infrared dataset for cross-modality person identification and discuss the differences among these datasets and ours. We also propose a novel modality augmentation method for cross-modality vision tasks. It can effectively reduce modality variance and integrated with other common data augmentation operations. In addition, we propose a novel multi-masking triplet loss to improve the traditional triplet, which ensure that hard positive sample is closer to easy positive sample and hard negative sample is far from anchor. Experimental results on three datasets demonstrate that our method achieves remarkable improvements compared to the current state-of-the-art methods. In addition, we verify the effectiveness of our augmentation strategy and loss function in other visible and thermal ReID and person detection method.

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

## A  APPENDIX

### A.1  DATASETS AND EXPERIMENTAL SETTINGS

**Datasets.** All of the experiments are performed on benchmark datasets SYSU-MM01 Wu et al. (2017) and RegDB Nguyen et al. (2017). The detailed introduction of dataset has been described in Sec. 2. In this paper, we evaluate our method on RegDB visible-to-thermal and thermal-to-visible modes and SYSU-MM01 by all search and indoor search single search. On our proposed NPU-ReID, we carried out extensive testings and adopting an integrated evaluation mode on the other two datasets.

**Evaluation metrics.** All experiments follow widely used standard evaluation protocols for Re-ID task. Cumulative Matching Characteristics (CMC) and mean Average Precision (mAP) are adopted

as the evaluation metrics. CMC(rank-r accuracy) is more concerned with the top-R positive sample in ranking results, and mAP is determined by the ranking results of all positive samples in the gallery.

**Experimental settings.** The proposed method is implemented in PyTorch. The improved dual-path Swin Transformer is adopted as the backbone. All models are trained on NVIDIA 3090Ti GPU and the pre-trained weights on ImageNet are used to initialize the model. A batch size of 8, an initial learning rate of 0.01, and the SGD optimizer with momentum 0.9 and weight decay 5e-4 are used.

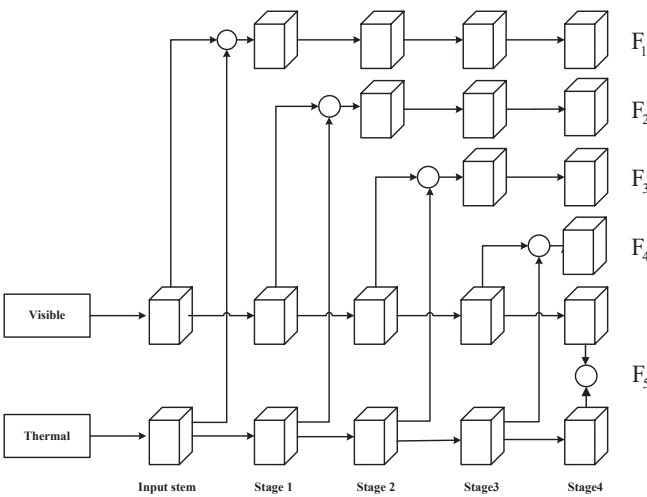

Figure 7: Illustration of network architecture with different fusion way.

## A.2 THE EVALUATION ON OTHER CROSS-MODALITY METHOD

**Applicability with Other ReID Methods** To assess whether MAG and MMT loss benefit other network, we incorporate our method into several popular representative network, such as TSLFN, AGW and CAJ. THe experimental results on two datasets are shown in Table 6. It is obviously that the network performance is all improved when utilizing the MAG augmentation and MMT loss function, which strongly demonstrates the effectiveness of the proposed method for cross-modality ReID tasks.

**Visible-Infrared Pedestrian Detection.** To demonstrate the generalizability of Modality augmentation, we also evaluate our proposed augmentations incorporated on cross-modality pedestrian detection task. We conduct the baseline method and the CFT method using CSPDarknet53 and CFB as backbone on LLVIP dataset and use the default settings with the authors' released code. The results in Table 7. demonstrate that the performance can be significantly improved under various metrics. Across these methods, our approach consistently improves the origin performance. Equipped with our strong augmentation, the performance further improves to 66.6%. It would be further improved by fine-tuning the hyper-parameters, such as the number and the scale limitation of exchanging area.

Table 6: Applicability of our proposed channel augmentation with state-of-the-art methods on NPU-ReID and RegDB dataset.

| Method | MAG+MMT | NPU-ReID | | | | RegDB | | | |
|---|---|---|---|---|---|---|---|---|---|
| | | R1 | R10 | R20 | mAP | R1 | R10 | R20 | mAP |
| TSLFN | | 83.78 | 98.99 | 99.74 | 69.10 | 86.65 | 96.94 | 98.50 | 72.73 |
| TSLFN | ✓ | **85.26** | **99.16** | **99.78** | **72.35** | **88.40** | **96.75** | **98.35** | **79.56** |
| AGW | | 79.77 | 98.37 | 99.52 | 68.36 | 70.05 | - | - | 66.37 |
| AGW | ✓ | **81.95** | **98.58** | **99.64** | **71.49** | **79.82** | **96.29** | **98.17** | **71.49** |
| CAJ | | 60.49 | 91.43 | 97.16 | 44.59 | 85.03 | 95.49 | 97.54 | 79.14 |
| CAJ | ✓ | **80.18** | **98.37** | **99.46** | **67.45** | **88.47** | **96.59** | **98.21** | **83.39** |

Table 7: Evaluation of our proposed modality augmentation incorporated on cross-modality pedestrian detection method.

| Model | Backbone | mAP50 | mAP75 | mAP |
|---|---|---|---|---|
| YOLOv5 | CSPDarknet53 | 95.8 | 71.4 | 62.3 |
| YOLOv5+MAG | CSPDarknet53 | **96.4** | **73.7** | **63.5** |
| CFT | CFB | 97.5 | 72.9 | 63.6 |
| CFT+MAG | CFB | **97.8** | **73.6** | **66.5** |

### A.3 THE EXPLORATION OF NETWORK ARCHITECTURE.

In this subsection, we explore how many parameters a dual-path transformer network should share, which is still not well investigated in the existing works.

We first explore which is the most reasonable structure of feature extractor. Several ablation experiments are conducted to explore the impact of share parameters on the performance of network. We build the following baseline network with Swin Transformer model. As is shown in Figure 7, The network is divided into input stem and four stages, and it is set to learn cross-modality features from stages 1, 2, 3 or 4 in sequence. We evaluate the performance of different backbone networks with different shared mechanisms on NPU-ReID datasets and RegDB datasets, as is listed in Table 8. We find that it is not effective to learn cross-modality features from beginning or learn only intra-modality features from beginning to end, such as F1 and F5. As we can see from the result of F3 shared mechanism, only after the network has fully learned the modality-specific features, modality-shared features can represent not only infrared information but also visible information, which is more beneficial for the subsequent tasks. The experiment results has verified the network of learning modality-specific features in the first two stages and learning modality-shared features in the last two stages can achieve the best performance.

Table 8: The results of different fusion architecture on RegDB and NPU-ReID.

| Structure | NPU-ReID | | | | RegDB | | | |
|---|---|---|---|---|---|---|---|---|
| | R1 | R10 | R20 | mAP | R1 | R10 | R20 | mAP |
| F1 | 79.93 | 96.24 | 98.38 | 70.04 | 83.5 | 94.81 | 97.67 | 78.73 |
| F2 | 81.96 | 98.44 | 99.12 | 72.29 | 83.54 | 95.19 | 98.01 | 78.23 |
| F3 | **82.36** | **98.52** | **99.85** | **73.57** | **87.89** | **95.73** | **98.12** | **84.91** |
| F4 | 81.6 | 98.47 | 98.89 | 72.8 | 83.11 | 92.3 | 95.58 | 81.34 |
| F5 | 80.23 | 97.63 | 98.32 | 71.16 | 85.8 | 94.11 | 97 | 82.75 |

