# OpenReview forum: "A NEW PARADIGM FOR CROSS-MODALITY PERSON RE-IDENTIFICATION"
_ICLR.cc/2023/Conference — Submitted to ICLR 2023_

### Official Review · Reviewer_hWin · 2022-10-21

**Confidence:** 4
**Correctness:** 3
**Technical Novelty And Significance:** 2
**Empirical Novelty And Significance:** Not applicable
**Recommendation:** 3

**Clarity, Quality, Novelty And Reproducibility:**

The current methodology section is somewhat confusing (especially Sec. 4.2). Modifications and clearer descriptions are needed. Grammatical errors are frequent, and the paper may require careful proofreading.
The newly constructed visible-infrared dataset has advantages over existing ones. On the other hand, the proposed method contains relatively incremental theoretical contributions compared to existing works.
Many hyperparameters used in experiments are not provided, which may increase the difficulty of reproduction.


**Details Of Ethics Concerns:**

The newly constructed dataset contains human subjects, but does not mention whether consent was obtained from the subjects.

**Strength And Weaknesses:**

Pros:
(1) A new larger-scale visible-infrared dataset is proposed, which is collected by more cameras and with a well-balanced amount of data for each modality.
(2) The proposed method achieves state-of-the-art performance in most experiments.

Cons:
(1) Eq. 1 appears to be inconsistent with the description in Sec. 4.2 and Fig. 5. From the description, both positive and negative samples seem to come from different modality from the anchor. Is there a labeling error in Eq. 1? If this is true, the proposed multi-mask triplet loss looks like a modified version of batch-hard triplet loss by changing the sampling strategy (i.e., restricting positive and negative samples to only one other modality). Furthermore, additional constraints on positive samples to make them closer are also often considered in other retrieval or matching tasks. In this case, the incremental modification somewhat limits the theoretical novelty of the proposed method.
On the other hand, this positive and negative sampling strategy is specialized for cross-modality matching. In real application scenarios, visible-infrared, visible-visible, and infrared-infrared matching may all be needed, while such a sampling strategy seems to only work in the visible-infrared case.

(2) More ablation experiments are needed to demonstrate the effectiveness of the proposed loss function and augmentation strategy. Specifically, the proposed multi-mask triplet loss should be compared with the general batch-hard and batch-all triplet loss, and the proposed augmentation, i.e., exchanging parts of the image with a positive sample from another modality, should be compared with the mentioned baseline strategy, i.e., changing image regions using random samples. In addition, there are several hyper-parameters in the proposed augmentation, such as the exchanging probability, area, and location. It is better to analyze their sensitivities.

(3) Lack of some related works on cross-modality person re-identification using two-stream frameworks. Some of the latest works have also achieved competitive or even better performance. More relevant papers need to be mentioned and differences need to be explained.
For examples:
- 1. Jiang et al. Cross-Modality Transformer for Visible-Infrared Person Re-Identification, ECCV 2022.
- 2. Zhang et al. FMCNet: Feature-Level Modality Compensation for Visible-Infrared Person Re-Identification, CVPR 2022.
- 3. Liu et al. Learning Memory-Augmented Unidirectional Metrics for Cross-modality Person Re-identification, CVPR 2022.
- 4. Zhang et al. Dual Mutual Learning for Cross-Modality Person Re-Identification, IEEE transactions on circuits and systems for video technology, Vol. 32, No. 8, 2022.
- 5. Chen et al. Structure-Aware Positional Transformer for Visible-Infrared Person Re-Identification, IEEE transactions on image processing, Vol. 31, 2022.
- 6. Zhang et al. Attend to the Difference: Cross-Modality Person Re-Identification via Contrastive Correlation, IEEE transactions on image processing, Vol. 30, 2021.
- 7. Lu et al. Cross-modality Person re-identification with Shared-Specific Feature Transfer, CVPR 2020.

(4) Minor:
- To avoid misleading, the results of the proposed method that are not the best in the tables should not be bolded.
- For easier understanding, it is better not to define too many variables that are never used in the following content (e.g., in Sec. 4.3). Useful variables should be defined in advance (e.g., $r$ and $A$).
- Some abbreviations need to be defined, e.g., MMAP and MMAN in Sec. 4.3.
- Some typos and grammatical errors exist, for example,
the sample strategy random sample -> the sampling strategy randomly samples (p. 6),
random selected -> randomly selected (p. 6),
compared -> compared to (p. 6),
resulting at the training is inefficient -> resulting in the training being inefficient (p. 6),
one identities -> one identity (p. 6).


**Summary Of The Paper:**

This paper presents a new visible-infrared dataset, NPU-ReID, for cross-modality person identification. Compared to existing datasets, this dataset contains more subjects, and the amount of visible and infrared data is well-balanced.
This paper also proposes a dual-path framework for cross-modality person re-identification, with Swin transformer as the backbone, combined with a modality augmentation strategy, and a multi-mask triplet loss to compress distances for sample pairs from different modalities. Experiments show the effectiveness of the proposed method.


**Summary Of The Review:**

The newly constructed visible-infrared dataset has advantages over existing ones, and may be beneficial for future cross-modality person re-id research. However, considering the limited theoretical novelty of the proposed method, the lack of experiments and references to related works, as well as the writing quality, I’m inclined to reject.

---

### Official Review · Reviewer_2Ybv · 2022-10-23

**Confidence:** 5
**Correctness:** 2
**Technical Novelty And Significance:** 2
**Empirical Novelty And Significance:** Not applicable
**Recommendation:** 3

**Clarity, Quality, Novelty And Reproducibility:**

The main ideas of the paper are not novel or have limited novelty. Some key resources (e.g., code, data) are unavailable which make it difficult to reproduce the main results.

**Strength And Weaknesses:**

Pros:
+ This paper explores the effectiveness of transformer model for cross-modality person re-identification, and presents a new cross-modal feature learning paradigm.

+ This paper builds a comprehensive visible-infrared dataset, which could assist the development community

Cons:
- The motivation and the idea of the MAG are nearly the same with [1] and [2] which reduces the modality gap by introducing semi-modality . The authors should clarify the difference and the advantage.
- The transformer model has also been explored in visible-infrared person re-identification by methods such as [3] and [4]. It is necessary to discuss and compare these methods.
- I feel confused why authors write the paper into a complicated style. It deeply makes it hard to understand paper. For example, the details of the loss function which is similar to [5] are not presented clearly enough. The logic of the introduction of the methods section is confusing.
- Some terms are confusingly defined and inconsistently expressed. For example, MMAP and MMAN in page 6; MMTP and MMTN in Table 2.
- The writing could be improved. For example, In calculation. The number of negative…

[1] Sfanet: A spectrum-aware feature augmentation network for visible-infrared person reidentification[J]. IEEE Transactions on Neural Networks and Learning Systems, 2021.

[2] Channel Augmented Joint Learning for Visible-Infrared Recognition. In ICCV 2021.

[3] Structure-Aware Positional Transformer for Visible-Infrared Person Re-Identification. In TIP 2021.

[4] CMTR: Cross-modality Transformer for Visible-infrared Person Re-identification[J]. arXiv preprint arXiv:2110.08994, 2021.

[5] Visible-Infrared Person Re-Identification via Homogeneous Augmented Tri-Modal Learning, in IEEE Transactions on Information Forensics and Security, vol. 16, pp. 728-739, 2021.


**Summary Of The Paper:**

This paper proposes a dual-path fusion network based on transformer blocks, and builds a visible-infrared dataset for cross-modality person re-identification.

**Summary Of The Review:**

This paper introduces a new dataset for visible-infrared person re-identification. However, this dataset does not provide additional insights for the community. It is just an another dataset. For the technical part, the novelty is too limited for ICLR. Seems that the presented details are already known in this field. The writing should also be polished. Therefore, the quality of this paper might not be suitable for ICLR.

---

### Official Review · Reviewer_heJg · 2022-10-24

**Confidence:** 4
**Clarity, Quality, Novelty And Reproducibility:** 1. The paper is poorly organized and …
**Correctness:** 3
**Technical Novelty And Significance:** 2
**Empirical Novelty And Significance:** 3
**Recommendation:** 3

**Strength And Weaknesses:**

Strengths:

	1. This paper presents a large-scale cross-modality Re-ID dataset .
	2. This paper presents an augmentation method that generates semi-modality sample by randomly exchanging patches between modalities. The experiments show the effectiveness of such augmentation for cross-modality person Re-ID.
	3. A multi-task triplet loss is designed for cross-modality person Re-ID.

Weaknesses:

	1. The paper is poorly organized. It is hard to quickly get the motivations and main ideas of the proposed methods.
	2. The thermal sensor and environment setting for data collection is not described in details. From Figure 2, why is the quality of thermal image significantly higher than RegDB and SYSU-MM01 ? Does the thermal sensor or capturing time cause it?
	3. The paper presents a transformer-based network as backbone,  what is the benefits over the CNN-based backbones in traditional methods? The reason using such a transformer-based method is not clearly discussed.
	4. The proposed multi-task triplet loss is not clearly clarified. It is strongly to re-organize the part and make a proof-reading. In addition, It seems there is mistake in Eq. (1). I suppose a max( x,0) is lost for both terms in $L_{mtri}$.
	5. The sensitivity of hyper-parameters such as $m_1$, $m_2$, $\lambda$ is not discussed. In particular, their values are not specified in the paper.
	6. There are lots of grammar mistakes, typos and description blurs that makes the paper hard to follow. It is strongly to find some experts to make a proofreading.


**Summary Of The Paper:**

This paper studies the cross-modality person re-identification problem and tries to build a new paradigm. Particularly, this paper proposes a large scale dataset called NPU-ReID that includes more identities and identities with even modality distribution. What's more, this paper presents 1) a modality augmentation method by synthesizing semi-modality samples and 2) a multi-mask triplet loss  by constraining both cross-modality triplet and intra-modality simple/hard positive pairs. Experiments show the effectiveness of the proposed method

**Summary Of The Review:**

From the discussion above, though the paper contributes a new dataset and raises some techniques to build a baseline, the novelty is incremental. What is more, the writing quality is quite low. It is far from a decent ICLR paper.

---

### Official Review · Reviewer_Wz3t · 2022-10-26

**Confidence:** 5
**Clarity, Quality, Novelty And Reproducibility:** See weaknesses.
**Correctness:** 2
**Technical Novelty And Significance:** 2
**Empirical Novelty And Significance:** 2
**Recommendation:** 3

**Strength And Weaknesses:**

Strength:
1. The authors build a new benchmark named NPU-ReID.
2. The authors propose a dual-path fusion network for cross-modal person re-identification.
3. The authors propose a modality data augmentation strategy and a cross modality triplet loss for optimization
4. The experiments are conducted on NPU ReID, RegDB and SYSU-MM01 datasets.

Weakness:
1. The novelty of the submission is limited for community. The authors only combine some existing techniques such as swin transformer, two stream feature fusion, modality triplet loss.
2. As stated in Table 1, there are no obvious advantages compared with SYSU-MM01 and the proposed NPU-ReID. Both of them contain hundred-level identities and similar numbers of images. The authors should clarifiy the advantages of NPU-ReID for community, compared with the previous datasets.
3. Do all the identities in NPU-ReID assign the agreement of privacy?  If the authors claim collecting and releasing a new dataset as a contribution of the submission, the dataset must be released for community after acceptance.
4. As shown in Figure 3, the authors use a two-stream network to fuse visible and infrared modalities. Are there any feature fusions between stage3 and stage4? If so, how to use the two-stream network for evalution? If not, the authors should clarify the detailed operations of stage 3 and stage 4 for training phase in Figure 3.
5. The authors use swin transformer and 224x224 input for experiments. It is not a fair comparison for previous work. The swin transformer is a stronger pretrained models than ResNet, which leads to better performance. Besides, previous methods usually use 256x128 input size.
6. The proposed data augmentation method is also limited. It is only used for the identities which contain both visible and infrared modality images. If there is no paired images for some identities, the proposed augmentation may not work.

**Summary Of The Paper:**

The authors focus on cross-modality person re-identification tasks. They build a new benchmark named NPU-ReID, and propose a dual-path fusion network. Besides, they also propose a modality data augmentation strategy and a cross modality triplet loss for optimizations. The experiments are conducted on NPU-ReID, RegDB and SYSU-MM01 datasets.

**Summary Of The Review:**

See weaknesses.

If the authors address all my concerns or point out that I misunderstand some parts of the submission, I am glad to improve the final rating.

---

### Decision · Program_Chairs · 2023-01-20

**Decision:**

Reject

**Justification For Why Not Higher Score:**

Reviewers have consistent negative ratings, and the authors did not provide responses to reviewers' comments.

**Justification For Why Not Lower Score:**

N/A

**Metareview: Summary, Strengths And Weaknesses:**

This paper aims to address the cross-modality person re-identification task and presents a new visible-infrared dataset named NPU-ReID. The authors proposed a modality augmentation strategy to leverage heterogeneous properties for the cross-modality person re-identification task, and conducted evaluations on the new datasets as well as existing benchmarks.

This paper has some meaningful contributions, such as the new dataset and the new augmentation method. The proposed cross-modal feature learning paradigm is also interesting.

Meanwhile, there are some major issues in the current version. First, the paper is poorly written and difficult to follow. Second, the novelty of proposed method is not well justified. Third, some related works are missing, and ablation studies are insufficient. The authors did not provide responses to the comments and questions from reviewers.

**Summary Of Ac-Reviewer Meeting:**

N/A